# Disparities by race and insurance-status in declines in pediatric ED utilization during the COVID19 pandemic

**Bisakha Pia Sen**[1]*, **Anne Brisendine**[1], **Nianlan Yang**[2], **Pallavi Ghosh**[3]

**1** Department of Health Care Organization & Policy, School of Public Health, University of Alabama at Birmingham (UAB), Birmingham, Alabama, United States of America, **2** SHP Research Collaborative, School of Health Professions, University of Alabama at Birmingham (UAB), Birmingham, Alabama, United States of America, **3** Division of Pediatric Emergency Medicine, School of Medicine, University of Alabama at Birmingham (UAB), Birmingham, Alabama, United States of America

☯ These authors contributed equally to this work.
* bsen@uab.edu

**Data Availability Statement:** The data (minimal anonymized data set and any additional data required to replicate the reported study finding in their entirety) has been made publically available (without any PHI): Your openICPSR Project

## Abstract

Pediatric Emergency Department (ED) utilization in the U.S. saw large declines during the COVID19 pandemic. What is relatively unexplored is whether the extent of declines differed by race and insurance status. An observational study was conducted using electronic medical record (EMR) data from the largest pediatric ED in Alabama for 2020 and 2019. The four subgroups of interest were African-American (AA), Non-Hispanic White (NHW), privately insured (PRIVATE), and publicly insured or self-insured (PUBLIC-SELF). Percentage changes in the 7-day moving average between dates in 2020 and 2019 were computed for total and high-severity ED visits by subgroup. Trends in percentage changes were plotted. T-tests were used to compare mean changes between subgroups. Large percentage declines in total ED visits and somewhat smaller percentage declines in high-severity visits were observed from March 2020. Declines were consistently larger for AA than NHW and for PUBLIC-SELF than PRIVATE. T-test results indicated mean date-specific percentage declines were significantly larger for AA than NHW for total visits (-38.92% [95% CI: -41.1, -36.8] versus -29.11% [95% CI: -30.8, -27.4]; p<0.001) and high-severity visits (-24.31% [95% CI: -26.2, -22.4] versus -19.49% [95% CI:-21.2, -17.8]; p<0.001), and larger for PUBLIC-SELF than PRIVATE for total visits (-36.32% [95% CI:-38.4, -34.3] versus 27.63% [95% CI:-29.2, -26.0]; p<0.001) and high-severity visits (-21.72% [95% CI: -23.5, -19.9] versus -20.01% [95% CI: -21.7, -18.3]; p = 0.04). In conclusion, significant differences by race and insurance status were observed in the decline in ED visits during the COVID19 pandemic, including high-severity visits. Minority-race and publicly insured or self-insured children often depend on the ED for health needs, lacking a usual source of care. Thus, these findings have worrisome implications regarding unmet healthcare needs and future exacerbations in health disparities.

openicpsr-152201: Published Access the deposit workspace at: https://www.openicpsr.org/openicpsr/workspace?goToPath=/openicpsr/152201&goToLevel=project View the published project: https://www.openicpsr.org/openicpsr/project/152201/version/V1/view.

**Funding:** This research was partially supported by an internal grant from the Office of the Dean, Lister Hill Center for Health Policy, and Sparkman Center for Global Health at the School of Public Health, UAB. No other external funding was received for this grant. The funders provided support in the form of salaries for the statistician for the project, Dr Nianlan Yang, and for graduate research assistants, but did not have any additional role in the study design, data collection and analysis, decision to publish, or preparation of the manuscript. While corresponding author Dr Bisakha Sen, and co-author Dr. Anne Brisendine are faculty in the Department of Health Care Organization & Policy, School of Public Health, UAB, they have no direct affiliation with the Office of the Dean, Lister Hill Center for Health Policy, or Sparkman Center for Global Health, and this does not impact the authors' adherence to PLOS ONE policies on sharing data and materials.

**Competing interests:** While corresponding author Dr Bisakha Sen, and co-author Dr. Anne Brisendine are faculty in the Department of Health Care Organization & Policy, School of Public Health, UAB, they have no direct affiliation with the Office of the Dean, Lister Hill Center for Health Policy, or Sparkman Center for Global Health, and this does not impact the authors' adherence to PLOS ONE policies on sharing data and materials.

## Introduction

It is well documented that pediatric emergency department (ED) utilization in the United States declined sharply during the first several months of the COVID19 pandemic [1–5]. What is relatively unexplored is whether the magnitude of these declines differed by indicators of socioeconomic disadvantage, like minoritized race or being low-income. Racial minorities and low-income communities bore the brunt of adverse health and economic consequences from the COVID19 pandemic [6–8]; hence, their healthcare utilization may have been especially disrupted. Further, the decline in pediatric ED visits has been studied using single ED and multi-state ED data, but the single ED studies have largely focused on Northeastern states [2, 9, 10], and relatively little information exists about what happened to pediatric ED visits in Deep South states which are characterized by large African-American populations, high poverty, and poor performance on health indicators [11–13] as well as higher rates of ED use [14, 15] compared to the rest of the nation pre-pandemic. Hence, information on pediatric ED utilization and disparities therein during the COVID19 pandemic in this region is critical from a public health perspective. Further, the low COVID19 vaccination rates in the Deep South states–for example, Alabama and Mississippi at the end of September 2021 had fully vaccinated 52.0% and 53.2% of their populations respectively compared to Pennsylvania with 68.5% or New York with 75.3% fully vaccinated [16, 17]–implies that these states may remain vulnerable to the disruptions caused by COVID19 for a longer period of time than states in other parts of the country. Therefore, this study fills an important gap in the literature by investigating changes in pediatric ED visits and disparities therein in a Deep South state during the COVID19 pandemic.

## Materials and methods

An observational study was conducted using data from the Children's of Alabama ED (CoA-ED)–the largest pediatric ED in the state, located in the city of Birmingham (Jefferson County). CoA serves as a teaching hospital for the University of Alabama at Birmingham (UAB) [18]. CoA-ED is Alabama's main tertiary care center for pediatric patients, the state's only designated Level 1 pediatric trauma center. It saw approximately 72,000 pediatric patients per year between 2015 and 2019. In order to compare pre- and peri-COVID19 pandemic ED visit data for a full 12-month period, we selected January 1—December 31, 2019 and 2020 for the study period.

Using electronic medical record (EMR) data for pediatric patients aged 0–18 years presenting to the CoA-ED during January 1—December 31, 2019 and 2020, information on date of visit, race-ethnicity, insurance status, acuity, and disposition were extracted. 'High-severity' visits were defined as visits meeting any of the following: (i) in the 2 highest levels of acuity (on the 5-point scale Emergency Severity Index (*ESI*) based on acuity and resource-needs [19]); (ii) categorized as 'trauma' visits; and (iii) visits that resulted in inpatient admission. The four subgroups of interest were African-American (AA), Non-Hispanic White (NHW), privately insured (PRIVATE), and publicly insured or self-insured (PUBLIC-SELF). Due to small sample sizes, no subgroups were created for Hispanic, 'other race', or unknown and out-of-state insurance status, and they were omitted from the analyses.

To examine how the volume of total and high-severity ED visits changed over 2020 compared to 2019 for each of the 4 subgroups of interest, the following approach was used. First, daily counts of total and high-severity visits were computed for all dates in 2019 and 2020. Next, 7-day moving averages (MA7) were constructed for each date in order to smooth the variations in ED visit volume across weekdays and weekends that have been recorded in existing literature [20–23] and better compare changes in volume of visits for each date in 2020

compared to the same date in 2019, where dates fall on different days of the week between the two years. Next, "date-specific percentage changes"–i.e., percentage changes in MA7 for each date in 2020 compared to its corresponding date in 2019 were computed, omitting February 29, 2020. Date-specific percentage changes by subgroups were plotted for total visits and high-severity visits over January 7—December 31, 2020, since using MA7 necessitated exclusion of the first 6 days of January. Two-sided t-tests (significance at P<0.05) were used to compare differences in mean date-specific percentage changes between AA and NHW and between PUBLIC-SELF and PRIVATE for the full range of dates as well as for pre- and post-March 13, 2020 (the date when the U.S. declared a national emergency in response to COVID19 and the first case was detected in Alabama). As part of sensitivity analyses, we also compared PRIVATE with just publicly insured children (Medicaid and CHIP) while omitting self-pay.

Furthermore, because AA pediatric patients are more likely to be publicly insured or self-insured than privately insured compared to NHW patients, thus making it difficult to discern whether insurance-status played a role in change in ED visits over and beyond the role played by race, we also inspected whether there were differences in how visits changed for PUBLIC--SELF versus PRIVATE patients within the racial categories of AA and NHW. To do this, we repeated the previous steps and computed percentage changes in MA7 in total visits for patients who were African-American and privately insured (AA_PRIVATE), African-American and public or self-insured (AA_PUBLIC-SELF), Non-Hispanic White patients who were privately insured (NHW_PRIVATE) and Non-Hispanic White patients who were publicly insured or self-insured patients (NHW_PUBLIC-SELF). As a final step, we conducted two-sided t-tests to compare differences in mean date-specific percentage changes for AA_PRIVATE versus AA_PUBLIC-SELF, and for NHW_PRIVATE versus NHW_PUBLIC-SELF.

Stata (v16) was used for all analyses; R(v3.6.2) was used for graphs. The study protocol was approved by the UAB Institutional Review Board for Human Use.

## Results

There was a combined total of 118,370 pediatric CoA-ED visits over 2019 and 2020 (Table 1), of which 28,504 met the definition of high-severity. Of these pooled total visits (high-severity visits), 48.9% (60.67%) were NHW; 49.5% (37.8%) were AA; 0.44% (0.34%) were Hispanic; and 1.2% (1.2%) were other race, biracial, or unknown. The last two groups were excluded from the analyses. Furthermore, 27.4% (33.2%) of total visits (high-severity visits) were PRIVATE and 72.2% (66.8%) were PUBLIC-SELF, which included 68.7% (64.7%) Medicaid or Children's Health Insurance Program (CHIP), 3.5% (1.8%) self-insured, 0.3% (0.3%) out-of-state, and 0.04% (0.02%) unknown insurance status. Again, these last two groups were excluded from the analyses. In further breakdowns not shown in the table, it was seen that, in the final pooled sample, of the children who were NHW, 41.2% were PRIVATE and 58.8% were PUBLIC-SELF (56.1% publicly insured, 2.7% were self-insured), whereas of the children who were AA, 12.7% were PRIVATE and 87.3% were PUBLIC-SELF (82.7% were publicly insured and 4.6% were self-insured).

Date-specific percentage changes in 2020 compared to 2019 for total and high-severity ED visits are presented in Fig 1, with '0' indicating no percentage change from one year to the next on the same date. Date-specific percentage changes for total visits for all groups became negative after March 13, 2020. The sharpest declines occurred in April and May of 2020, with greater than 70% declines for AA and PUBLIC-SELF and almost 60% declines for NHW and PRIVATE. A partial rebound happened from June to September 2020, especially for NHW and PRIVATE, which were just 10–20% lower than 2019 numbers, though visits for AA and PUBLIC-SELF were 40–50% lower. Percentage declines became steeper again after September

**Table 1. Distribution of patient socioeconomic characteristics for pooled pediatric ED visits for Jan-Dec 2019 and 2020, Children's of Alabama Emergency Department.**

| | Total Visits | | High-Severity Visits | |
|---|---|---|---|---|
| | N = 118,370 | | N = 28,504 | |
| Race and Ethnicity | N | Percent | N | Percent |
| Non-Hispanic White | 57907 | 48.92% | 17282 | 60.63% |
| African-American | 58581 | 49.49% | 10775 | 37.80% |
| Hispanic [a] | 521 | 0.44% | 97 | 0.34% |
| Other Race/Biracial/Unknown [a] | 1420 | 1.20% | 351 | 1.23% |
| Insurance Status | | | | |
| Public (Medicaid or CHIP) | 81320 | 68.70% | 18445 | 64.71% |
| Self-Pay | 4143 | 3.50% | 505 | 1.77% |
| Private | 32493 | 27.45% | 9475 | 33.24% |
| Out of State [a] | 367 | 0.31% | 86 | 0.30% |
| Unknown [a] | 47 | 0.04% | 6 | 0.02% |
| Gender | | | | |
| Male | 62073 | 52.44% | 15130 | 53.08% |
| Female | 56297 | 47.56% | 13374 | 46.92% |
| Age-range | | | | |
| 0–3 years | 49112 | 41.49% | 10048 | 35.25% |
| 3<-6 years | 17590 | 14.86% | 3190 | 11.19% |
| 6<-12 years | 27770 | 23.46% | 6944 | 24.36% |
| 12<-18 years | 23899 | 20.19% | 8323 | 29.20% |

[a] The data are based on EMRs from Children's of Alabama Emergency Department, the largest ED in the state. Due to small sample sizes, these sub-groups were excluded from the final analyses.

2020 and were particularly steep in November and December 2020. Percentage declines were consistently larger for AA compared to NHW, and for PUBLIC-SELF compared to PRIVATE.

It can also be seen from Fig 1 that date-specific percentage changes for high-severity visits also became negative after March 13, 2020, though the magnitudes of decline were smaller than total visits. There were brief and transient rebounds in the summer and fall of 2020, when the volume of visits matched or exceeded those of 2019, but they mostly showed a decline. While differences in declines between groups were less discernible for high-severity visits compared to total visits, they appeared to be larger for AA than NHW and for PUBLIC-SELF than PRIVATE.

T-test results (Table 2) indicated mean date-specific percentage declines in 2020 compared to 2019 were significantly larger for AA than NHW for total visits (-38.92% [95% CI: -41.1, -36.8] versus -29.11% [95% CI: -30.8, -27.4], p<0.001) as well as for high-severity visits (-24.31% [95% CI: -26.2, -22.4] versus -19.49% [95% CI:-21.2, -17.8], p<0.001). T-test results also indicate that mean date-specific percentage declines were larger for PUBLIC-SELF than PRIVATE for total visits (-36.32% [95% CI:-38.4, -34.3] versus 27.63% [95% CI:-29.2, -26.0], p<0.001) and for high-severity visits (-21.72% [95% CI: -23.5, -19.9] versus -20.01% [95% CI: -21.7, -18.3], p = 0.04). The differences in declines were more significant after March 13, 2020 (-46.37% for AA, -34.88% for NHW, p<0.001 for total visits; -28.91% for AA, -23.69% for NHW, p<0.001 for high-severity visits; -43.46% for PUBLIC-SELF, -32.87% for PRIVATE, p<0.001 for total visits, -26.61% for PUBLIC-SELF, -23.54% PRIVATE, p = 0.006 for high-severity visits). In contrast, mean date-specific percentage changes prior to March 13, 2020 were small, and differences between AA and NHW, and PUBLIC-SELF and PRIVATE largely

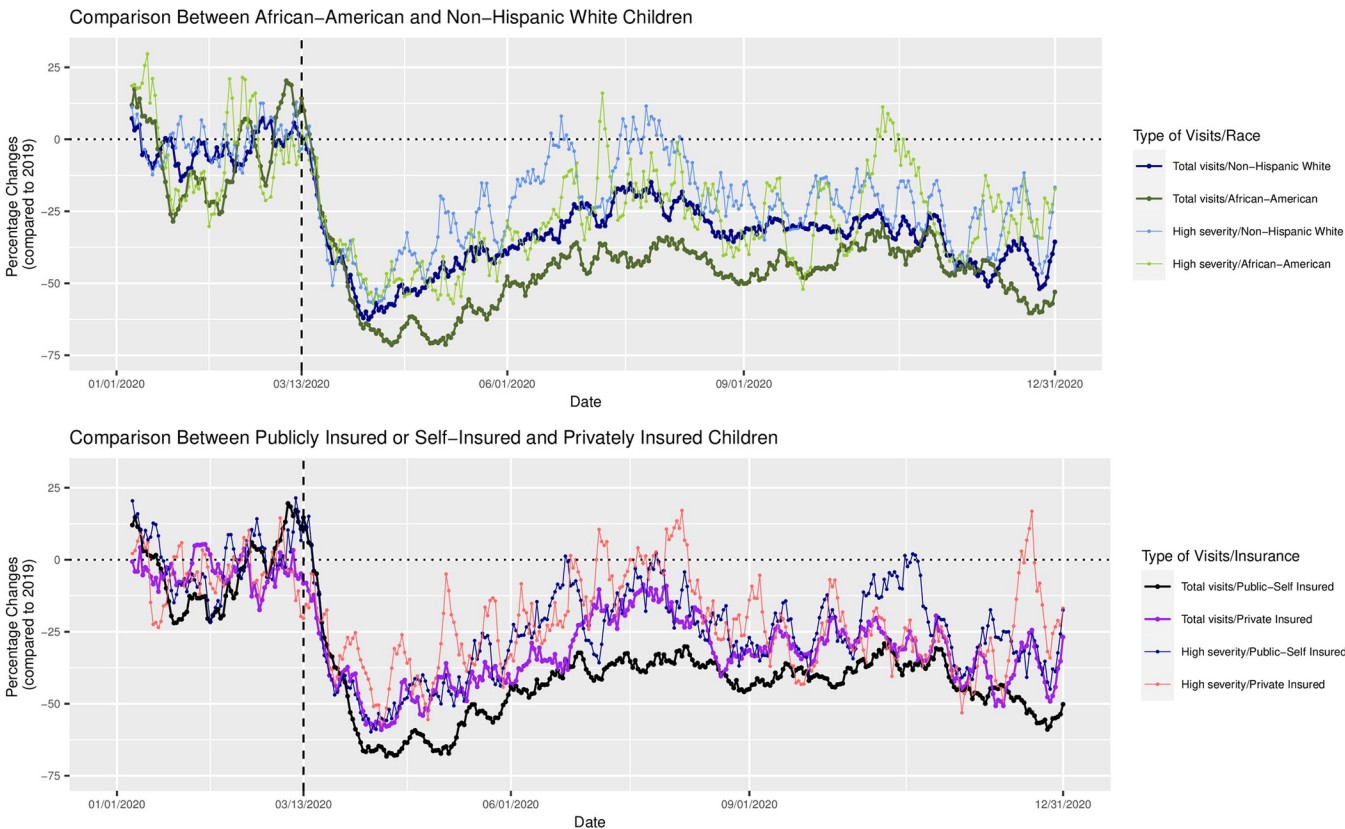

**Fig 1. Date−specific percentage changes of 7−day moving average on ED visits in total and high-severity ED visits in 2020 compared to 2019.**

**Table 2. Differences in the mean date-specific percentage changes in 2020 compared to 2019 for total and high-severity ED visit counts by race and insurance status.**

| | Mean Percentage Change in Total ED Visits Between 2019 and 2020 [95% C.I.] | | | | | |
|---|---|---|---|---|---|---|
| | African-American Children | Non-Hispanic White Children | P-value from T-test | Publicly Insured or Self-Insured Children | Privately Insured Children | P-value from T-test |
| Jan 1-Dec 31, 2020 compared to 2019 | -38.92 [-41.1, -36.8] | -29.11 [-30.8, -27.4] | P<0.001 | -36.32 [-38.4, -34.3] | -27.63 [29.2, -26.0] | P<0.001 |
| Jan 1-Mar 13, 2020 compared to 2019 | -5.24 [-8.8, -1.7] | -3.01 [-4.3, -1.7] | P = 0.182 | -3.98 [-7.0,-1.0] | -3.90 [-5.1,-2.7] | P = 0.904 |
| Mar 13-Dec 31, 2020 compared to 2019 | -46.37 [-47.8, -44.9] | -34.88 [-36.2, -33.6] | P<0.001 | -43.48 [-44.9,-42.0] | -32.87 [-34.2,-31.6] | P<0.001 |
| | Mean Percentage Change in High-Severity ED Visits Between 2019 and 2020 | | | | | |
| | African-American Children | Non-Hispanic White Children | P-value from T-test | Publicly Insured or Self-Insured Children | Privately Insured Children | P-value from T-test |
| Jan 1-Dec 31, 2020 compared to 2019 | -24.31 [-26.2, -22.4] | -19.49 [-21.2, -17.8] | P<0.001 | -21.72 [-23.5,-19.9] | -20.01 [-21.7,-18.3] | P = 0.04 |
| Jan 1- Mar 13, 2020 compared to 2019 | -3.51 [-7.4, 0.4] | -0.52 [-2.1, 1.1] | P = 0.186 | 0.18 [-2.3,-2.7] | -4.01 [-6.1, -1.8] | P = 0.016 |
| Mar 13-Dec 31, 2020 compared to 2019 | -28.91 [-30.7, -27.2] | -23.69 [-25.4, -22.0] | P<0.001 | -26.56 [-28.3, -24.9] | -23.54 [-25.3, 21.8] | P = 0.001 |

Notes: Date-specific percentage changes were computed by calculating differences in 7-day moving averages for 2020 from the 7-day moving averages of the corresponding dates in 2019. Hispanic, "other race", out-of-state insurance, and unknown insurance were not analyzed due to small sample size.

**Table 3. Differences in the mean date-specific percentage changes in 2020 compared to 2019 for total ED visit counts between privately insured & publicly/self-insured children, by race.**

| | Mean Percentage Change in Total ED Visits Between 2019 and 2020 [95% C.I.] | | | | | |
| | African-American Children | | | Non-Hispanic White Children | | |
| | Publicly Insured or Self-Insured Children | Privately Insured Children | P-value from T-test | Publicly Insured or Self-Insured Children | Privately Insured Children | P-value from T-test |
|---|---|---|---|---|---|---|
| Jan 1-Dec 31, 2020 compared to 2019 | -40.07 [-42.3, -37.9] | -34.15 [-36.5, -31.8] | P<0.001 | -31.95 [-33.9, -29.9] | -25.95 [-27.6, -24.3] | P<0.001 |
| Jan 1-Mar 13, 2020 compared to 2019 | -6.82 [-10.6, -3.0] | -1.82 [-6.7, 3.1] | P = 0.03 | -1.87 [-4.7, 1.0] | -4.10 [-5.9, -2.3] | P = 0.28 |
| Mar 13-Dec 31, 2020 compared to 2019 | -47.10 [-48.7, -45.5] | -40.99 [-42.8, -39.2] | P<0.001 | -38.31 [-39.8, -36.8] | -30.57 [-31.9, -29.2] | P<0.001 |

Notes: For each sub-group, date-specific percentage changes were computed by calculating differences in 7-day moving averages for 2020 from the 7-day moving averages of the corresponding dates in 2019.

insignificant. Sensitivity analyses comparing PRIVATE with just publicly insured children yielded virtually identical results.

T-test results comparing mean date-specific percentage changes in total visits for AA_PRIVATE versus AA_PUBLIC-SELF, and for NHW_PRIVATE versus NHW_PUBLIC-SELF are shown in Table 3. These results indicated that mean date-specific percentage declines in 2020 compared to 2019 were significantly larger for AA_PUBLIC-SELF than AA_PRIVATE (-40.07% [95% CI: -42.3, -37.9] compared to -34.15% [95% CI: -36.5, -31.8], p<0.001, and were also significantly larger for NHW_PUBLIC-SELF than NHW_PRIVATE (-31.95% [95% CI: -33.9, -29.9] compared to -25.95% [95% CI: -27.6, -24.3], p<0.001). As with previous results, the declines were larger and the differences in declines were more significant after March 13, 2020.

## Discussion

A growing literature has documented declines in pediatric ED visits in the U.S. during the first several months of the COVID19 pandemic [1–5]. However, the question of whether there were disparities in these declines by race and insurance status is relatively unexplored, beyond two single-ED studies that considered racial differences in changes in ED visits for mental health [9, 10]. The primary aim of this study was to address this gap and investigate differences in declines for children in the AA, NHW, PUBLIC-SELF, and PRIVATE groups. A secondary aim was to compare declines for children in the PUBLIC-SELF and PRIVATE subgroups within the categories of AA and NHW. Results showed that, during March to December 2020, relative declines in total ED visits as well as high-severity ED visits were larger for children in the AA and PUBLIC-SELF groups than for children in the NHW and PRIVATE groups. Declines were also significantly higher for the PUBLIC-SELF subgroup compared to PRIVATE subgroups for both the categories AA and children.

This paper's findings of steep declines in pediatric ED visits starting in March 2020, followed by a partial rebound from June to September 2020, and then a subsequent decline after September 2020, are consistent with existing literature (S1 Fig, Adjemian, 2021, https://stacks.cdc.gov/view/cdc/104808) [4]. However, separating the analyses by groups reveals that the rebound was mostly for children in the PRIVATE and NHW groups. This is important from a public health perspective. AA and economically-disadvantaged communities, who are likely to be publicly insured or self-insured, disproportionately use the ED for pediatric healthcare needs [24], as they often lack access to a usual source of care. Given that the larger declines

were seen for high-severity visits as well, it is possible that urgent healthcare needs went unmet for disadvantaged children.

This study also contributed by focusing on a Deep South state. Most single-ED studies on this topic area have focused on Northeastern states [2, 9, 10], and one multi-state study that looked at race-ethnicity as a variable of interest found only a small decline in the share of AA patients–from 22.3% in pre-COVID19 years to 21.4% in 2020 [1]. This is markedly different from this study's finding, and it underscores the possibility that multi-state studies will mask region-specific variations in decline in healthcare utilization during the pandemic. Hence, greater focus on trends in states with large shares of disadvantaged pediatric populations is warranted.

The data used in this study does not permit deciphering whether the decline in ED visits portends unmet healthcare needs or whether it happened because of reduced incidence of illness and injury due to school closures and stay at home ordinances. However, along with the greater decline in ED visits found in this study for children in the AA and PUBLIC-SELF groups, there is evidence of relatively greater declines in pediatric vaccination in Michigan among publicly-insured children [25], worsening racial disparities in pediatric obesity in Pennsylvania [26], and–despite reports of worsening pediatric mental health during the pandemic [27]–disproportionate declines in pediatric ED visits for mental health for AA children than NHW children in Connecticut [9] and Pennsylvania [10]. Finally, preliminary (unpublished) findings from a pilot project by this team suggests that AA and low-income children were less likely to access telehealth than their NHW and higher income counterparts; therefore, it is unlikely that care provided via telehealth compensated for the declines in ED visits for these disadvantaged groups. Taken in conjunction, these findings strongly indicate that essential healthcare was foregone by minority and low-income communities, and disparities in pediatric health conditions likely worsened during the COVID19 pandemic.

This study has certain limitations. First, the information was derived from EMR data; there was no information on reasons why patients did not come to the ED, whether care was being accessed in alternate settings such as via urgent care or office visits, or whether there were subsequent adverse health consequences of foregone or delayed care. Second, though the Hispanic community has been hard-hit by the COVID19 pandemic, the sample of Hispanic patients using the CoA-ED was too small to permit meaningful examination of changes in their ED use. Third, the EMR data received from CoA-ED did not include ICD-10 codes for diagnosis or claims, so this study could not document or tabulate conditions with which patients were presenting; thus, it cannot be deciphered how many pediatric patients were presenting with COVID19 infections, particularly in the latter part of 2020. Finally, this is a single-ED study, which limits generalizability but, since Alabama shares several socio-demographic and economic characteristics with other Deep South states [11–13, 28], findings from the largest pediatric ED in Alabama are likely to be pertinent for those states as well.

## Conclusion

This study found that, in a part of the country that already performs poorly in health and socio-economic metrics, declines in pediatric ED visits were disproportionately and persistently larger for AA children compared NHW children, and for economically disadvantaged children–as indicated by publicly insured or self-insured status–compared to privately-insured children. These differences in declines were apparent for total visits and for high-severity visits. Given the relatively slow uptake of COVID19 vaccines in this region of the country, and the continued risk that new variants of the virus can emerge, it is likely that disruptions to healthcare utilization caused by the pandemic may persist in the near future. Hence, surveillance by

healthcare systems and providers is urgently called for to detect and minimize future health consequences from foregone pediatric ED care, particularly among disadvantaged pediatric populations.

## Author Contributions

**Conceptualization:** Bisakha Pia Sen, Anne Brisendine.

**Formal analysis:** Bisakha Pia Sen.

**Funding acquisition:** Bisakha Pia Sen.

**Methodology:** Nianlan Yang.

**Resources:** Pallavi Ghosh.

**Supervision:** Pallavi Ghosh.

**Writing – original draft:** Bisakha Pia Sen.

**Writing – review & editing:** Anne Brisendine, Pallavi Ghosh.

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
