## [Decision Letter · Decision Letter 0]

24 Sep 2021

PONE-D-21-26799Disparities by Race and Insurance-Status in Declines in Pediatric ED Utilization During the COVID-19 Pandemic.PLOS ONE

Dear Dr. Sen,

Thank you for submitting your manuscript to PLOS ONE. After careful consideration, we feel that it has merit but does not fully meet PLOS ONE’s publication criteria as it currently stands. Therefore, we invite you to submit a revised version of the manuscript that addresses the points raised during the review process.

We look forward to receiving your revised manuscript.

Kind regards,

Jingjing Qian

Academic Editor

PLOS ONE

Journal Requirements:

 [The authors were partially supported through an internal grant by the School of Public Health, University of Alabama at Birmingham. The funders had no role in study design, data collection and analysis, decision to publish, or preparation of the manuscript.]

Please respond by return e-mail so that we can amend your financial disclosure and competing interests on your behalf.

4. Thank you for providing the following Funding Statement:  

[The authors were partially supported through an internal grant by the School of Public Health, University of Alabama at Birmingham. The funders had no role in study design, data collection and analysis, decision to publish, or preparation of the manuscript.]. 

We note that one or more of the authors is affiliated with the funding organization, indicating the funder may have had some role in the design, data collection, analysis or preparation of your manuscript for publication; in other words, the funder played an indirect role through the participation of the co-authors. 

If the funding organization did not play a role in the study design, data collection and analysis, decision to publish, or preparation of the manuscript and only provided financial support in the form of authors' salaries and/or research materials, please review your statements relating to the author contributions, and ensure you have specifically and accurately indicated the role(s) that these authors had in your study in the Author Contributions section of the online submission form. Please make any necessary amendments directly within this section of the online submission form.  Please also update your Funding Statement to include the following statement: “The funder provided support in the form of salaries for authors [insert relevant initials], but did not have any additional role in the study design, data collection and analysis, decision to publish, or preparation of the manuscript. The specific roles of these authors are articulated in the ‘author contributions’ section.” 

If the funding organization did have an additional role, please state and explain that role within your Funding Statement. 

Please also provide an updated Competing Interests Statement declaring this commercial affiliation along with any other relevant declarations relating to employment, consultancy, patents, products in development, or marketed products, etc.  

6. Please amend your list of authors on the manuscript to ensure that each author is linked to an affiliation. Authors’ affiliations should reflect the institution where the work was done (if authors moved subsequently, you can also list the new affiliation stating “current affiliation:….” as necessary).

Reviewers' comments:

Reviewer's Responses to Questions

**Comments to the Author**

1. Is the manuscript technically sound, and do the data support the conclusions?

Reviewer #1: Partly

Reviewer #2: Yes

2. Has the statistical analysis been performed appropriately and rigorously? 

Reviewer #1: No

Reviewer #2: Yes

3. Have the authors made all data underlying the findings in their manuscript fully available?

Reviewer #1: No

Reviewer #2: Yes

4. Is the manuscript presented in an intelligible fashion and written in standard English?

Reviewer #1: Yes

Reviewer #2: Yes

5. Review Comments to the Author

Reviewer #1: This study evaluates the trends in pediatric emergency department visits in 2019 and 2020 (during the COVID-19 pandemic). The authors evaluate these trends in subgroups based on race, ethnicity, and insurance status. In addition, the study addresses an important topic given the well-known racial disparities further revealed by COVID, especially in the southern United States. Some issues with the methods and results reporting need to be clarified/addressed. Please see below for more specific comments by section:

Introduction:

• More references are needed to support a few of the statements made in the introduction.

o The 4th sentence ending in “…higher rates of ED use compared to the rest of the nation pre-pandemic” needs more support.

o Additionally, the penultimate sentence in that section needs more detail regarding vaccination rates and disparities specific to the region, perhaps as it compares to the Northeastern states mentioned earlier.

Methods:

• Study site: Please provide some more detail about the study site including annual patient volume, tertiary/teaching, etc. to help the audience draw conclusions regarding generalizability of the results.

• Study dates: I suspect the study periods were chosen to compare pre- and peri-pandemic data in a full 12 month period, however, please clarify rationale for selecting these dates.

• While the variables provided are laid out clearly, some more detail is necessary to develop a better picture of the dataset. If available, some more demographic data would be helpful to compare the 2019 and 2020 groups, specifically age.

• Please clarify further the 5-point acuity scale – is this ESI?

• Smoothing: Please provide some references for rationale.

Results:

• Throughout this section, please specify the years for each month mentioned for consistency and clarification.

• Consider combining the third and fourth paragraphs of this section to better incorporate the numbers (paragraph 4) alongside the text (paragraph 3).

• Table 1 – Clearly presented overall. Please provide the raw numbers alongside the percentages.

• Figure 1 – Please include the years on each graph. Per the title of the figure, the figures are comparing 2020 and 2019, but this should be clarified with labels.

• Table 2 – Very important table! Please clarify dates in the first column (All 12 months, up to March 13, after March 13) with years and specific time periods (e.g. January 1, 2020-December 31, 2020; January 1, 2020 – March 13, 2020; March 14, 2020-December 31, 2020). Additionally, please report 95% confidence intervals in this table. Worth highlighting this in the text (including abstract).

Discussion:

• The discussion here is interesting and keeps with the paper’s focus. The authors have done a good job addressing all the major points. However, parts of the discussion would benefit from some more detail and support from the literature.

• The following statements should be discussed in more detail and/or supported with findings from the literature:

o The first sentence in the discussion – “Growing literature has documented declines…” – please cite the literature here.

o Supplementary Figure 1 by Adjemian et al. was not included in the document.

o Please expand on inability to document specific health conditions. As the authors mention, this is an important limitation.

o The sentence – “…since Alabama shares several socio-demographic and economic characteristics with other Deep South states” – please support this statement as well.

Additionally, there are some minor grammatical and typographic errors (i.e. U.S. instead of United States on first use in the introduction) – please check for these throughout the document.

Reviewer #2: The authors set out to explore the differences in the utilization of the ED at a major pediatric emergency department in Alabama during the Covid-19 pandemic based on racial and socioeconomic factors. The authors found a significantly larger decline in ED utilization by the African-American and publicly insured or self-insured patients in comparison to non-Hispanic white and privately insured patients, respectively. These findings inform the reader of the further worsening of healthcare disparities in the pandemic environment and suggest an exacerbation of unmet healthcare needs of vulnerable populations.

1. Parts of the manuscript are somewhat difficult to read. I recommend reviewing the revised manuscript for grammar and syntax errors.

2. It appears that the authors chose to group publicly- and self-insured patients together. What was the breakdown of those groups? Are those populations truly comparable enough to be considered together?

3. The age range of the patients was not defined. Was there a difference among groups? This could be another variable relevant for analysis.

4. Was there an overlap in the two sets of groups (AA and NHW and between PUBLIC-SELF and

PRIVATE)? In other words, for instance, was there a difference in the AA + PUBLIC-SELF and AA + PRIVATE? Is possible that the difference is more related to the race or the insurance status?

5. Results Section, paragraph 2, last sentence: the statement is somewhat ambiguous, as it appears to imply that the groups were combined (AA and PUBLIC vs. NHW and PRIVATE)

6. Page 7, Discussion, sentence starting with: While this paper’s findings of steep declines… is unclear the way it is written.

7. The authors make a great point discussing the lower rates of utilization of telemedicine among African American and low income children. However, is there any data on urgent care use? Is it possible that the people were more likely to take their children to urgent care facilities due to shorter lines and more predictable wait times, especially for low-severity visits?

6. PLOS authors have the option to publish the peer review history of their article (what does this mean?). If published, this will include your full peer review and any attached files.

Reviewer #1: No

Reviewer #2: No

---

## [Author Response · Author response to Decision Letter 0]

27 Oct 2021

We have uploaded a separate file that includes details of how we responded to each comment by the editor and reviewers. We want to draw specific attention to the funding statement which was an issue that had been raised by the editor (this is also in our Response file, but deserves highlighting). Our new funding statement reads as follows:

"This research was partially supported by an internal grant from the Office of the Dean, Lister Hill Center for Health Policy, and Sparkman Center for Global Health at the School of Public Health, UAB. No other external funding was received for this grant. The funders provided support in the form of salaries for the statistician for the project, Dr Nianlan Yang, and for graduate research assistants, but did not have any additional role in the study design, data collection and analysis, decision to publish, or preparation of the manuscript. While corresponding author Dr Bisakha Sen, and co-author Dr. Anne Brisendine are faculty in the Department of Health Care Organization & Policy, School of Public Health, UAB, they have no direct affiliation with the Office of the Dean, Lister Hill Center for Health Policy, or Sparkman Center for Global Health, and this does not impact the authors’ adherence to PLOS ONE policies on sharing data and materials. Dr Pallavi Ghosh is attending physician at CoA-ED, from where the data for this research were obtained. CoA-ED had no role in the research beyond providing the data, and this also does not impact the authors’ adherence to PLOS ONE policies on sharing data and materials."

---

## [Decision Letter · Decision Letter 1]

15 Nov 2021

PONE-D-21-26799R1Disparities by Race and Insurance-Status in Declines in Pediatric ED Utilization During the COVID-19 Pandemic.PLOS ONE

Dear Dr. Sen,

Thank you for submitting your manuscript to PLOS ONE. After careful consideration, we feel that it has merit but does not fully meet PLOS ONE’s publication criteria as it currently stands. Therefore, we invite you to submit a revised version of the manuscript that addresses the points raised during the review process.

We look forward to receiving your revised manuscript.

Kind regards,

Jingjing Qian

Academic Editor

PLOS ONE

Journal Requirements:

Additional Editor Comments:

Thanks the authors for addressing most of the reviewers' comments. In the next revision, please:

1) edit text to clarify the comparisons between AA and NHW, and between Public-self and Private throughout to avoid any confusions as suggested by Reviewer 2. For example, the last sentence in the 2nd paragraph of the Results section, "Percentage declines were consistently larger for AA and PUBLIC-SELF than NHW and PRIVATE." This is confusing and you did not revise as suggested by Reviewer 2. This is one example but please read the entire manuscript to improve clarify of writing.

2) the author's response to Reviewer 2's question regarding the overlap between AA+Public-self and AA+Private is not convincing. Please provide results for the direct comparison between these 2 subgroups in table and in text.

3) please giving the manuscript another read through for grammar and syntax prior to final submission as suggested by Reviewer 1.

Reviewers' comments:

Reviewer's Responses to Questions

**Comments to the Author**

1. If the authors have adequately addressed your comments raised in a previous round of review and you feel that this manuscript is now acceptable for publication, you may indicate that here to bypass the “Comments to the Author” section, enter your conflict of interest statement in the “Confidential to Editor” section, and submit your "Accept" recommendation.

Reviewer #1: All comments have been addressed

2. Is the manuscript technically sound, and do the data support the conclusions?

Reviewer #1: Yes

3. Has the statistical analysis been performed appropriately and rigorously? 

Reviewer #1: (No Response)

4. Have the authors made all data underlying the findings in their manuscript fully available?

Reviewer #1: Yes

5. Is the manuscript presented in an intelligible fashion and written in standard English?

Reviewer #1: Yes

6. Review Comments to the Author

Reviewer #1: The authors have addressed the initial concerns. I would recommend giving the manuscript another read through for grammar and syntax prior to final submission.

7. PLOS authors have the option to publish the peer review history of their article (what does this mean?). If published, this will include your full peer review and any attached files.

Reviewer #1: No

---

## [Author Response · Author response to Decision Letter 1]

30 Nov 2021

We have uploaded a file that details how we responded to the remaining concerns from editor and reviewers. Please let us know if anything further is needed.

---

## [Editor Report · Decision Letter 2]

27 Dec 2021

Disparities by Race and Insurance-Status in Declines in Pediatric ED Utilization During the COVID-19 Pandemic.

PONE-D-21-26799R2

Dear Dr. Sen,

We’re pleased to inform you that your manuscript has been judged scientifically suitable for publication and will be formally accepted for publication once it meets all outstanding technical requirements.

Kind regards,

Jingjing Qian

Academic Editor

PLOS ONE

Additional Editor Comments (optional):

Thank you!
---

## [Editor Report · Acceptance letter]

17 Jan 2022

PONE-D-21-26799R2 

Disparities by race and insurance-status in declines in pediatric ED utilization during the COVID19 pandemic 

Dear Dr. Sen:

I'm pleased to inform you that your manuscript has been deemed suitable for publication in PLOS ONE. Congratulations! Your manuscript is now with our production department. 

Kind regards, 

on behalf of

Dr. Jingjing Qian 

Academic Editor

PLOS ONE